# Design Optimization of Stacked Pallet Load Units

**Piotr Sawicki** [1,*] and **Hanna Sawicka** [2]

1   Division of Transport Systems, Poznan University of Technology, ul. Piotrowo 3, 61-138 Poznań, Poland
2   Division of Rail Transport, Poznan University of Technology, ul. Piotrowo 3, 61-138 Poznań, Poland
*   Correspondence: piotr.sawicki@put.poznan.pl

**Abstract:** The article deals with the problem of building stacked pallet load units consisting of at least two stackable pallet load units. Moreover, this article concerns the part of the flow of goods in distribution networks that is prepared at the place of initial assembly in the form of palletized loading units designed for the final receiver. Such a unit does not exceed the limits of permissible weight or height. The article proposes a single-criteria binary programming model in which the goal is to minimize the pallet spaces required to accommodate the constructed units. In addition to the classical parameters of acceptable weight and height of the units, the constraints also take into account the fragility of the goods placed on each unit, filling the top layer of each unit, and its height homogeneity. The model developed was verified on a test dataset, and the savings from the use of optimum construction of the stacked palletized cargo units were demonstrated through the conducted experiments.

**Keywords:** sustainable logistics; sustainable transportation issues; pallet stacking; stacked pallet load unit problem (SPLUP)

## 1. Introduction

### 1.1. The Essence of Constructing Pallet Load Units

Forming palletized load units—PLU is an issue closely related to the functioning of distribution networks and supply chains. It concerns the proper placement of products on the smallest possible number of carriers. In practice, there are two basic types of palletized load units, i.e., homogeneous and heterogeneous. In the first case, products of one type are loaded on a pallet, which is mostly performed by product manufacturers. In the other case, the loaded products differ in basic characteristics such as weight, length, width and height, and product type, and this case mainly occurs at distributors and dealers.

The arrangement of products in PLUs has a number of practical consequences. Firstly, it affects the stability of the pallet units, which, on the one hand, may involve the risk of product damage during basic logistics operations such as transportation, storage, and handling. Product safety depends on their placement on the pallet as they need to be layered to prevent crushing the ones located on lower and intermediate layers. Secondly, product arrangement affects the cost of logistics operations. The smaller the number of PLUs, the less space necessary for transportation and storage, the less energy required, and the lower the cost of performing these operations. The comprehensive research on the distribution network structure and its effectiveness related to the type of packages, including PLUs and parcels are presented in the previous work of the authors; see Sawicki and Sawicka [1].

In general, the essence of the problem of forming palletized load units involves the proper arrangement of products on the pallet to ensure its stability and compliance with basic constraints such as the limited weight and height of the load unit. An overview of the current state-of-the-art in this area, including the categories of decision-making problems, the way these problems are modeled, the procedures for solving them as well as the results obtained, are presented in the following subsection.

Based on the results of the literature review and the authors' empirical experience, this publication focuses on the issue of forming a specific type of load unit, which is a stacked pallet load unit—SPLU, commonly referred to as a sandwich-pallet. This issue involves creating a single load unit composed of several elementary PLUs previously arranged on separate pallets. The need to build this type of unit is a direct result of the increasingly noticeable phenomenon within the distribution network of the formation of PLUs whose weight and/or height significantly deviate from the limit values, i.e., the available space on the pallets is not fully utilized. This results from the requirements of purchasers, mainly large retail chains, who order large volumes of goods versus a variety of products and expect that the PLUs will be prepared in such a way that the goods reaching their transshipment warehouses will not require any additional operations, except for the redirection of dedicated PLUs for shipment to the final receivers. Thus, creating a few collective units instead of many smaller PLUs is becoming a necessity. In practice, such operations are undertaken on an ongoing basis, i.e., whilst loading, the forklift operator subjectively assesses whether it is feasible to combine the available PLUs into SPLUs. The FLT operator uses his own experience and intuition, yet whether such an operation is correct, both in terms of the durability of product packaging on individual PLUs and the created weight and height of the SPLU, cannot be guaranteed. The work undertaken in this area by the authors of this article provides a proposal to solve this problem by developing and verifying a procedure to optimize the process of creating SPLUs.

*1.2. The State-of-the-Art Design of Palletized Load Units*

In the literature, the problem of planning pallet load units is called *bin packing problem—BPP*. To solve the problem, one needs to define which items of different sizes need to be packed into a finite number of bins or containers, each of a given and fixed capacity so as to minimize the number of bins used. In the literature, the same problem, depending on the authors, is referred to as pallet loading problem—PLP, e.g., Dell'Amico and Magnani [2], Morabito et al. [3], pallet building problem—PBP, e.g., Alonso et al. [4], container loading problem—CLP, e.g., Lim and Zhang [5], or packing problem—PP, e.g., Ali et al. [6].

Since the packing problem is an NP-hard decision problem, it can also be analyzed, with the exception of its principal constraint, i.e., the weight of the product, as a soft version of two dimensions (2D), i.e., width x length of the products to be packed, e.g., G and Kang [7], or a more complex problem of three dimensions (3D), i.e., width x length x height, e.g., Dell'Amico and Magnani [1], Morabito et al. [3], or Ali et al. [6].

From the perspective of the location, the packing problem is analyzed and planned with respect to homogenous items by some of the researchers who refer to it as manufacturer's pallet loading—MPL, e.g., Marabito et al. [3]. When packaging involves heterogeneous items, it is called distributor's pallet loading—DPL; see Akkaya et al. [8].

In general, the bin packing problem can be analyzed as a stand-alone packing problem or as a part of a more complex problem, i.e., in combination with vehicle loading. As far as the stand-alone approach is concerned, it has been extensively analyzed as a decision problem for around 50 years. In recent years several papers have been published to review or compare different approaches to this problem [6,9]. Silva et al. [9] reviewed the papers with respect to the methods proposed for the solution of the problem. In conclusion, a group of the most challenging methods was identified. Ali et al. [6] analyzed different approaches to the decision-making problem concerning packing. They differentiated 3D off-line vs. on-line streams of packing problems. Off-line packing problems can happen when full knowledge about items is available beforehand. The on-line problem, i.e., real-time problem, is when items arrive one by one, and the packing decision should be made immediately without prior knowledge about the items. According to Ali et al., most of the packing problems described in the literature are of the off-line type. Contemporary examples of the on-line bin packing problem can be found in Lin et al. [10], where a pattern-based adaptive heuristics for the on-line bin packing problem is proposed. The distribution

of items may be predicted based on the packed items, and the pattern is next applied in the packing of the items that arrive later.

Some new research into the bin packing problem was published in the last few years. Gzara et al. [11] analyzed a wide spectrum of practical constraints, including vertical support, load bearing, planogram sequencing, and weight limits. The authors performed extensive numerical tests to prove the ability of the approach to find high-quality solutions for industrial-size instances within a short computational time. There are also some recent papers on the application of more efficient computational procedures whose aim is to achieve computation results in the shortest time possible. Tresca et al. [12] published a paper on the 3D bin packing problem (3D-BPP) where they proposed a model oriented on pallets configurations to satisfy the practical requirements of item grouping by logistic features such as load bearing, stability, height homogeneity, overhang as well as weight limits, and robotized layer picking. The complex problem is solved with metaheuristics, which combines MILP formulation and layer-building heuristics. Moreover, 3D-BPP is analyzed in the work of Zuo et al. [13]. The authors addressed a novel 3D-BPP variant in which the shape-changing factor of non-rectangular and deformable items was incorporated into the model. In the research of Elhedhli et al. [14], another practical variant of 3D-BPP is addressed. The authors analyzed the mixed-case palletization problem, where item support with the presence of different sizes of items is considered. Elhedhli et al. proposed a novel problem formulation as well as a column-generation solution approach. El-Ashmawi and Elminaam [15] concentrated on the design of an approximate algorithm of BPP applicable to solve large-scale instances within a reasonable time. They proposed a modified version of the squirrel search algorithm (SSA) for solving the 1D bin packing problem. In the experiments, hard class instances of up to 200 items were tested, and the obtained result was compared against other approximate algorithms such as particle swarm optimization (PSO), African buffalo optimization (ABO), and crow search algorithm (CSA).

A typical complex decision problem is usually a combination of packing and vehicle loading problems [3,4,15]. In the work of Morabito et al. [3], the optimal solution to the problem is presented. However, the mathematical model is not presented in detail. This concept is applied to solve the combined problem of pallet and vehicle loading, and the size of unit packages and the size of pallets and vehicles are also discussed. A two-phase approach with a packing problem is also considered by Moura and Bortfeld [16]. In this approach, however, the main objective is to guarantee sufficient utilization of the truck loading space. Alonso et al. [4] considered the problem of building and placing pallets on the truck at the same time, i.e., pallet loading as the first phase and truck loading as the second phase. The authors implemented several extended constraints, including the total weight of the load, the maximum weight supported by each vehicle's axle, and the distribution of the load inside the vehicle. During the truck loading phase, it was allowed to stack one pallet on top of another. The model was constructed and solved with the GRASP algorithm, and the experiments were performed in the domain of the number of instances, vehicles, and computation time. Dell'Amico and Magnani [2] have also proposed a two-phase procedure. In the first phase, 2D layers were defined, while in the second phase, the layers were combined on the minimum number of pallets. The authors constructed a specialized metaheuristic with a MILP model of a 3D problem, which was subsequently solved by the Gurobi solver. The experiments were performed in the domain of the number of instances and computation time. Contrary to the previous research on BPP and vehicle loading problems, Moura Santos et al. [17] proposed the research on BPP with compatible categories, i.e., products that cannot be transported together. In this approach, the bin is the fleet of vehicles, and the problem is to allocate products with respect to the product categories. Moura Santos et al. concentrated on large instances of 200–1000 items, and they solved them with a variable neighborhood search (VNS) procedure. The experiments were also compared with alternative procedures.

There are several conclusions resulting from the state-of-the-art of the bin packing problem that should be highlighted, i.e.,

- Numerous papers on BPP focus on the analysis of a wide spectrum of constraints that are either theoretical or practical. There is hardly any research on small-size pallets, i.e., substantially below the maximum height or weight of the PLU. It is a typical situation in contemporary logistics.
- The BPP is an NP-hard decision problem, which is why a substantial number of studies have been carried out on the effectiveness of the optimization procedures.
- The case when PLUs can be stacked one on top of another is referred to as a combined problem of pallet and vehicle loading. It is not, however, a widely discussed topic in the literature.
- One specific way of formulating the BPP problem can be regarded as the inspiration for the formulation of the problem of stacking pallet load units, i.e., every single PLU can be treated as equivalent to a single layer considered in the PLU packing planning process.

### 1.3. Formulation of the Decision Problem

The article considers the problem of building stacked pallet load units (SPLUs) composed of elementary pallet load units (PLUs) whose height and gross weight parameters are significantly below the permissible values. Therefore, a mechanism is sought to plan the stacking of a set of PLUs in a way that will reduce the amount of space required on the load bed or in racking slots, while the parameters of the SPLUs built in this way should be within the permissible values of weight and height. It is assumed that the elementary PLUs are created in advance as a result of the adaptation to the specific requirements of the ordering party and may not be subject to content modification. It is also assumed that individual PLUs may contain homogeneous or mixed products, and the ordering party agrees to stack PLUs to form SPLUs.

The basic characteristics parameterizing the susceptibility of elementary PLUs to the formation of SPLUs include such parameters as gross weight and gross height, i.e., including the pallet itself, the fragility class of the stacked products, and the status of the PLUs resulting from the filling of its upper layers. Figure 1a,b show elementary PLUs with a total height well below the limit value; the PLUs in Figure 1a have a base status on which it is possible to stack more PLUs, and in Figure 1b, a *top* status: on this type of unit it is impossible to stack more PLUs. Figure 1c,d show SPLUs built from elementary PLUs. In the case of (c), all PLUs have *base* status, and in the case of (d), the highest layer is a *top*-type PLU, which makes it impossible to build further layers on it.

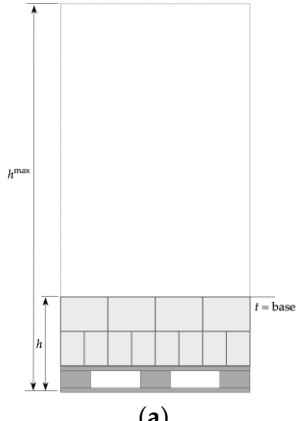
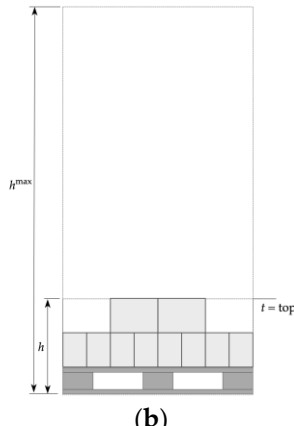
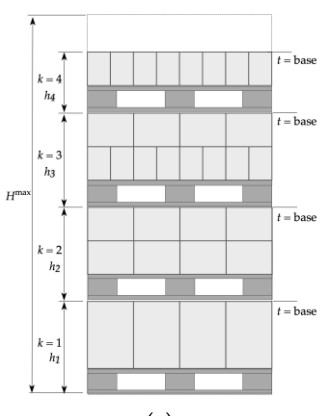
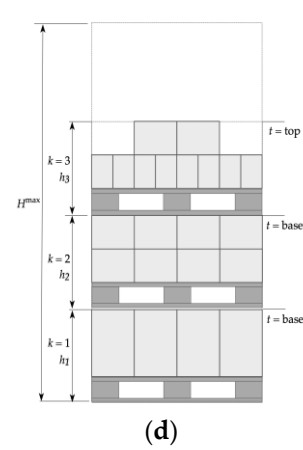

| (a) | (b) | (c) | (d) |

**Figure 1.** The considered forms of PLUs: (**a**) Elementary PLU of the *base* type, which has the necessary features for the construction of SPLUs; (**b**) Elementary PLU of the *top* type, which has limited susceptibility for the construction of SPLUs; (**c**) SPLU built only from PLUs of the *base* type; (**d**) SPLU built from SPLs of the *base* and *top* type.

*1.4. Purpose and Scope of the Research*

Based on the conclusions resulting from the state-of-the-art, see Section 1.2, and based on the formulation of the research problem, see Section 1.3, the purpose of the work was defined. The objective of the study is to develop a procedure to optimize the construction process of stacked pallet load units (SPLUs) built from elementary PLUs, taking into account, in addition to the classical limitations of the weight and height of the unit, the susceptibility of PLUs to the construction of successive layers, including the fragility of products, the flatness of the top layer of products contained on the PLU, and the filling of the top layer.

The article contains four key sections. In Section 1, the need for PLU formation as a cost driver in transportation and warehousing processes is discussed, the decision problem is defined, and the research objective is formulated. Section 2 presents the research methodology, including the definition of the objective function and a set of constraints; it also presents the assumptions for the proposed methodology, i.e., its applicability limits. Section 3 presents the results of computational experiments; the methodology is verified based on a test dataset. Section 4 is a summary of the article, where the obtained results are described with reference to the formulated purpose of the work, and the directions for further work are defined. Finally, a list of the literature used is presented.

## 2. The Research Methodology

*2.1. Notations*

In the presented research, some indexes, decision variables, and parameters are defined. The complete list is presented in the following tables; see Tables 1–3.

**Table 1.** The list of indexes in alphabetical order.

| Symbol | Definition |
|--------|------------|
| $i$ | an ordinal number of pallet load unit to be stacked; $i = 1, 2, \ldots, I$; |
| $j$ | an ordinal number of stacked pallet load unit; $j = 1, 2, \ldots, J$; |
| $k$ | an order of pallet load unit setting in the built stacked pallet load unit; $k = 1, 2, \ldots, K$, i.e., $k = 1$ is the lowest layer, and $k = K$ is the top layer. |

**Table 2.** The list of variables in alphabetical order.

| Symbol | Definition |
|--------|------------|
| $x_{ijk}$ | a binary variable specifying which $i$-PLU should be assigned to the $k$-layer on the $j$-SPLU. |
| $y_j$ | a binary variable specifying whether the $j$-SPLU is formed; |

**Table 3.** The list of parameters in alphabetical order.

| Symbol | Definition |
|--------|------------|
| $f_i$ | a fragility class of the products on the $i$-PLU (-), $f = 1, 2, \ldots, n$ |
| $h_i$ | a gross height of $i$-PLU, including height of pallet expressed in (mm) |
| $H_j^{\max}$ | the maximum height of $j$-SPLU, expressed in (mm) |
| $t_i$ | a parameter defining the upper layer of $i$-PLU (-); $t_i \in \{0; 1\}$, where: $t_i = 0$ if upper layer is flat and its filling is higher than 70% (*base* status); $t_i = 1$ otherwise (*top* status). |
| $w_i$ | a gross weight of $i$-PLU, including weight of pallet, expressed in (kg) |
| $W_j^{\max}$ | the maximum weight of $j$-SPLU, expressed in (kg) |

## 2.2. Key Assumptions

The following assumptions are made in the decision problem:

- Individual *i*-PLUs are built according to the final receiver's order. They contain one type of product, i.e., homogeneous PLU, as well as a mix of products, i.e., heterogeneous PLU, the so-called mix.
- In the case of heterogeneous *i*-PLU, containing products with different fragility classes, a representative fragility class is determined, i.e., the lowest fragility class among the products on the *i*-PLU.
- If the top layer of *i*-PLU is flat, that is, the total height of products accumulated on each layer of *i*-PLU is the same, then such a PLU can serve as the base for the construction of *j*-SPLU and its intermediate layers.
- If there is a difference in the height of the products from the point of view of the last layer on the PLU, i.e., the top layer is not flat, then such an *i*-PLU can only be the last layer of the built SPLU. The height of such an *i*-PLU is the maximum value for all the accumulated products on the *i*-PLU.
- If the last layer of *i*-PLU is filled up to at least 70%, such a PLU can serve as a base for the construction of SPLU and its intermediate layers.
- When the last layer of *i*-PLU is filled up to less than 70%, then such a unit can only be the last layer of the built SPLU.

## 2.3. Decision Variables and Objective Function

The mathematical model of the considered decision problem is formulated as a binary model and is presented in this section. The minimized objective function $F$ is the number of *j*-SPLU necessary to handle all elementary *i*-PLUs included in the order. It is represented by the following formulas, see (1)–(11):

$$F = \min \sum_{j=1}^{J} y_j \tag{1}$$

where

$$y_j = \begin{cases} 1 \text{ if } \sum_{i=1}^{I} \sum_{k=1}^{K} x_{ijk} > 0 \\ 0 \text{ if } \sum_{i=1}^{I} \sum_{k=1}^{K} x_{ijk} = 0 \end{cases}, \forall j = 1, 2, \ldots, J \tag{2}$$

The unit of *j*-SPLU can be planned if *i*-PLU is assigned at any of the available *k*-layers of *j*-SPLU. Otherwise, the *j*-SPLU unit is not planned.

## 2.4. Constraints

The optimization model of the design the stacked pallet load units problem is constructed with nine constraints; see Formulas (3)–(11) below. Constraint (3) indicates that each *i*-PLU is assigned exactly to one of the *j*-SPLU within the considered order. The next constraint (4) indicates that on a *k*-layer of the *j*-SPLU, only one *i*-PLU can be located. Two other constraints say that the gross weight (5) and gross height (6) of the *j*-SPLU result from the requirements of the *j*-SPLU user, i.e., $W_j^{\max}$ and $H_j^{\max}$. Based on the constraint (7), successive layers on a *j*-SPLU are planned while preserving the fragility class of individual *i*-PLU on a *k*-layer. It means that the *k*-layer may contain an *i*-PLU of the same or higher fragility class than layer $k-1$. Fragility class $f = 1$ means the lowest fragility, i.e., the product has a significant compression resistance, while $f = n$ means the highest fragility, i.e., the product has the lowest compression resistance. The constraints (8) and (9) result from the classification of *i*-PLUs into the *base* and *top* status. Status *base* means that on *i*-PLU, another *k*-layer of *j*-SPLU can be considered, and status *top*, otherwise. The unit *i*-PLU of the status *top* can be on the top of *j*-SPLU, exclusively. Constraint (8) guarantees that, at maximum, one *i*-PLU of *top* status can be allocated to a single *j*-SPLU. However, constraint (9) ensures

that the $i$-PLU of the *top* status will not be located below the $i$-PLU of the *base* status. The last two constraints, i.e., (10) and (11), indicate that both decision variables $y_j$ and $x_{ijk}$ are of binary nature.

$$\sum_{j=1}^{J} \sum_{k=1}^{K} x_{ijk} = 1, \forall i = 1, 2, \ldots, I \tag{3}$$

$$\sum_{i=1}^{I} x_{ijk} \leq 1, \forall j = 1, 2, \ldots, J; \forall k = 1, 2, \ldots, K \tag{4}$$

$$\sum_{i=1}^{I} \sum_{k=1}^{K} w_i \cdot x_{ijk} \leq W_j^{\max}, \forall j = 1, 2, \ldots, J \tag{5}$$

$$\sum_{i=1}^{I} \sum_{k=1}^{K} h_i \cdot x_{ijk} \leq H_j^{\max}, \forall j = 1, 2, \ldots, J \tag{6}$$

$$\sum_{i=1}^{I} f_i \cdot \left( x_{ijk} - x_{ij(k-1)} \right) \geq 0, \forall j = 1, 2, \ldots, J; \forall k = 2, \ldots, K \tag{7}$$

$$\sum_{k=1}^{K} \sum_{i=1}^{I} t_i \cdot \left( x_{ijk} - x_{ij(k-1)} \right) \leq 1, \forall j = 1, 2, \ldots, J \tag{8}$$

$$\sum_{i=1}^{I} t_i \cdot \left( x_{ijk} - x_{ij(k-1)} \right) \geq 0, \forall j = 1, 2, \ldots, J; \forall k = 2, \ldots, K \tag{9}$$

$$x_{ijk} = \begin{cases} 1 & \text{if } i\text{-PLU is located at } k\text{-layer in the } j\text{-SPLU}, \\ 0 & \text{otherwise} \end{cases} \tag{10}$$

$$y_j = \begin{cases} 1 & \text{if } j\text{-SPLU is planned to be designed}, \\ 0 & \text{otherwise} \end{cases} \tag{11}$$

Fragility is the parameter that defines the compressive strength of a single package of a product exerted by products of the same type and piled on it. Products, usually homogenous, with a certain fragility of a full load unit, i.e., filled up to the maximum weight, are characterized by the height of this PLU, i.e., $h_i^{\max}$. It should also be assumed that the weight of a full pallet load unit of products with lower fragility $f_i$ is substantially heavier than the weight of a full pallet load unit of products with higher fragility $f_{i+1}$, i.e., $w_i(f_i) > w_{i+1}(f_{i+1})$. Thus, in case of stacking incomplete pallet load units, i.e., $h_i < h_i^{\max}$, with different fragilities and maintaining the fragility ranking from the lowest to the highest $f_i > f_{i+1} > f_{i+2}$ etc., starting from the background of SPLU, the total height $H_j^{\max}$ of this SPLU may exceed the maximum height of each of the pallet full load unit of products, i.e., $H_j^{\max} > h_i^{\max}$.

### 2.5. Implementation Procedure

The optimization of the design of the stacked pallet load units is based on the formal assumptions and the mathematical model presented in Sections 3.1 and 3.2. The practical application of this approach is shown as a BPMN notation procedure, see Figure 2.

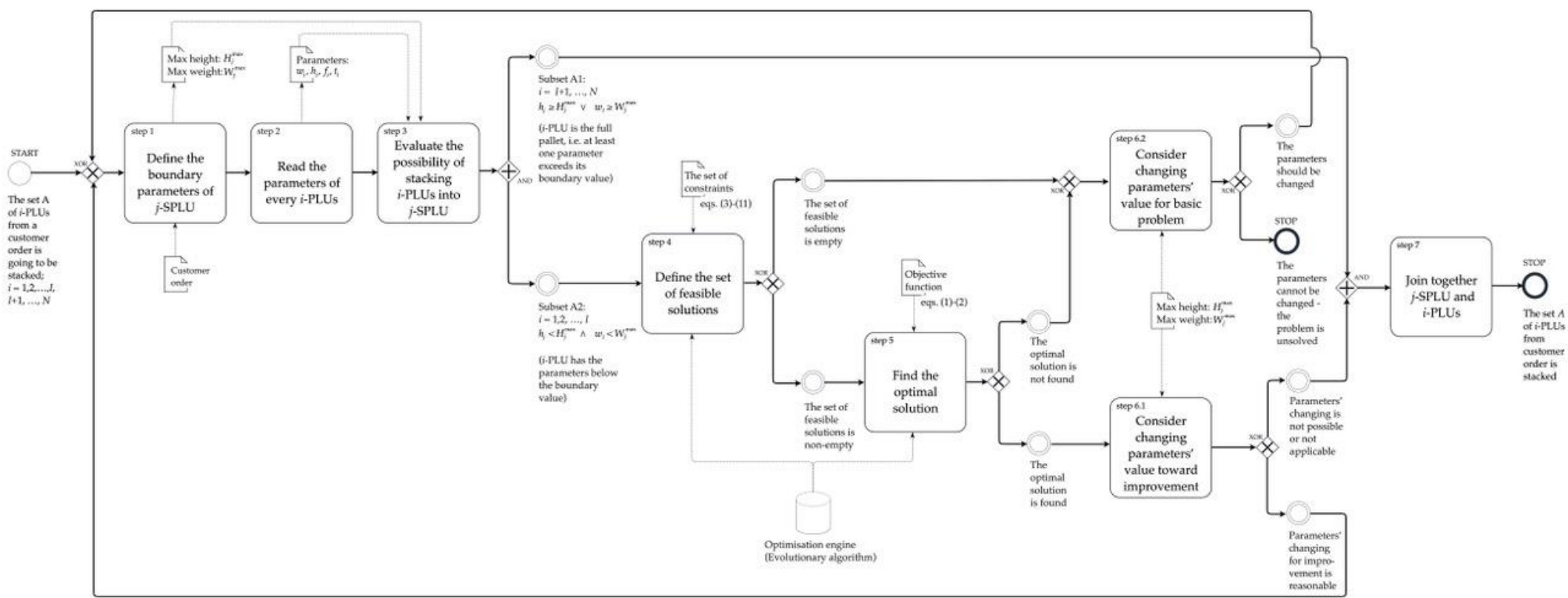

**Figure 2.** The implementation of stacking pallet load units optimization procedure.

The starting point is set $A$ of $i$-PLUs, where $i = 1, 2, \ldots, I, I + 1, \ldots, N$, which constitutes all pallets prepared upon customer order. Initially, two boundary parameters of $j$-SPLU that should be designed on $A$, i.e., $W_j^{\max}$ and $H_j^{\max}$, are defined (step 1). Then, each $i$-PLUs from the set $A$: $i = 1, \ldots, N$ is identified in terms of basic parameters (step 2), such as weight $w_i$, height $h_i$, fragility class $f_i$, and the degree of filling of the top layer $t_i$. Based on the identified parameters of each $i$-PLU and the boundary parameters of SPLU to be constructed, in the next step of the procedure (step 3), the assessment of the possibility of building stacked $j$-SPLUs from available $i$-PLUs is carried out. As a result, set $A$ is divided into two subsets, i.e., $A1$: $i = I + 1, \ldots, N$, which includes $i$-PLUs that do not meet at least one boundary condition of $W_j^{\max}$ or $H_j^{\max}$, and subset $A2$: $i = 1, 2, \ldots, I$, which includes those $i$-PLUs that do not exceed any of the boundary parameters. In step 4, a set of feasible solutions is built on $A2$; for this purpose, all the constraints described by Formulas (3)–(11) are applied. If, as a result of this operation, the set of feasible solutions is non-empty, in step 5, the optimal solution is searched; at this stage, the Formulas (1)–(2) of the objective function are applied. The computations within steps 4 and 5 are performed by optimization engines of the evolutionary algorithm. If, as a result of step 5, an optimal solution is found, in step 6.1, an analysis of whether it is necessary or applicable to modify boundary parameters for further improvement should be carried out. If not, then step 7 is performed, which consists of joining the $j$-SPLUs constructed in steps 4 and 5 with the $i$-PLUs that were separated in step 3 due to unmet boundary conditions; see subset $A1$ of step 3. The result of the procedure at this stage is a reduced number of pallet spaces that are necessary for the transport or storage of $i$-PLUs because some of them can be transformed to SPLUs, upon boundary conditions of $W_j^{\max}$ or $H_j^{\max}$.

If, as a result of step 6.1, the change of boundary parameters is applicable, then it is necessary to return to step 1 and repeat the optimization procedure step by step. If, after step 4, it is not possible to define a set of feasible solutions, or in step 5, it is not possible to find an optimal solution, it is recommended to consider changing the values of the boundary parameters (step 6.2) and repeat the procedure from step 1. If such a change of parameters is not possible, the problem is unsolved, and the procedure should be stopped.

## 3. Results of Computational Experiments

### 3.1. Assumptions, Input Data, and Computational Experiments

The computations verifying the methodology proposed and presented in Section 2 were carried out on a MacBook Pro computer with a 3.1 GHz dual-core Intel Core processor and a Windows 10 operating system embedded in a virtual machine. The model was implemented into the Excel environment, and the computations were performed using Frontline Solvers' evolutionary optimization engines.

In the experiments, 10 sample instances were analyzed, i.e., sets consisting of 8–20 PLUs, as per customer order, containing only the units to be potentially expanded into SPLUs, see subset $A2$ on Figure 2. The complete data sets of analysed instances are attached to this article, see Tables A1–A3 in Appendix A. SPLUs were then built from the elementary PLUs based on the mathematical model presented in Section 2, Equations (1)–(11). The results obtained are shown below in Table 4. The analysis included such parameters as the following:

- Minimum and maximum values: weight, height, and number of PLUs of *top* and *base* type;
- The structure of storage carriers, including PLU *top* and *base* types;
- Boundary parameters of height and weight of built SPLUs.

The results were analyzed taking into account the following:

- The number of SPLUs constructed; the value of the decision variable $y_j$;
- The degree of filling the available height and weight of SPLUs;
- Computing time to solve the problem.

**Table 4.** Results of computational experiments.

| Instance | Parameters [1] | | | | | Results [2] | | | |
|---|---|---|---|---|---|---|---|---|---|
| | Nº PLU | min $w_i$; $h_i$ | max $w_i$; $h_i$ | $t_i$ [3] | $f_i$ [4] | Nº SPLU | % $H^{max}$ | % $W^{max}$ | CT |
| 1 | 9 | 186; 311 | 256; 416 | 1; 8 | 1; 1; 7; 0; 0 | 3 | 91.0–95.3 | 78.1–82.2 | 0.7 |
| 2 | 9 | 175; 295 | 259; 421 | 2; 7 | 5; 3; 1; 0; 0 | 3 | 83.0–92.9 | 70.4–79.6 | 0.01 |
| 3 | 8 | 236; 405 | 361; 530 | 3; 5 | 1; 4; 3; 0; 0 | 4 | 77.6–83.0 | 69.8–77.4 | 0.8 |
| 4 | 8 | 245; 414 | 349; 518 | 2; 6 | 2; 3; 3; 0; 0 | 4 | 72.8–82.0 | 63.0–76.0 | 1.8 |
| 5 | 20 | 130; 300 | 390; 520 | 7; 13 | 2; 5; 4; 2; 7 | 9 | 40.9–99.2 | 41.9–86.6 | 51.2 |
| 6 | 20 | 182; 353 | 246; 433 | 5; 15 | 1; 7; 8; 4; 0 | 7 | 69.6–100.0 | 55.0–77.6 | 2.6 |
| 7 | 20 | 175; 344 | 341; 533 | 5; 15 | 6; 3; 2; 6; 3 | 8 | 80.7–99.8 | 70.1–78.7 | 1.6 |
| 8 | 17 | 148; 359 | 273; 442 | 5; 12 | 7; 10; 0; 0; 0 | 6 | 70.8–99.6 | 55.5–73.0 | 2.7 |
| 9 | 15 | 129; 347 | 266; 438 | 2; 13 | 6; 6; 3; 0; 0 | 5 | 90.6–99.6 | 53.7–72.9 | 1.6 |
| 10 | 19 | 225; 444 | 267; 571 | 7; 12 | 8; 3; 0; 3; 5 | 10 | 47.5–90.3 | 31.4–60.6 | 1.6 |

[1] Parameters are expressed in: Nº PJŁ (units), $w$ (kg), $h$ (mm), $t_i$ (-), and $f$ (-). [2] Results are expressed in: Nº SPLU (units), % $H^{max}$ (%), % $W^{max}$ (%), and CT (s). [3] Number of PLUs of *top*; *base* status. [4] Number of PLUs of fragility class 1; 2; 3; 4; 5.

### 3.2. Results and Discussion

The results of the experiments presented in Table 4 allow for the following conclusions and observations:

- The application of the proposed mathematical model allows for a significant reduction of the space required to locate PLUs in the transportation and/or warehousing process, thus creating fewer required pallet positions. In the analyzed instances, 3–10 SPLUs were created from 8–20 PLUs, which means a two- to three-times reduction in the demand for pallet positions in logistics processes. A smaller number of PLUs translates directly into lower transportation costs in logistics networks, as well as less demand for rack slots in the process of storing PLUs.
- The smaller the number of *top* status PLUs in the structure of stacked units, the better the chance of building a smaller number of SPLUs while maintaining all the constraints. In instances 1, 2, and 9 with at most two PLUs *top* type, the number of SPLUs is three times less than the number of PLUs; in the other instances, the higher number of *top* type PLUs results in at most 2–2.5 times reduction of the number of SPLUs compared to PLUs.
- The higher the values of weight and height of individual PLUs subjected to stacking, the fewer multi-layered SPLUs can be built. The structure of the SPLUs built depends on the specificity of the items located on the individual PLUs. In all instances (1–10), a higher degree of achievement of the max height parameter (82.0–100%) is obtained compared to the max weight parameter (60.6–86.6%). This means that the analyzed instances constitute a set of relatively light but hight products.
- The computing time to solve the problem of optimal stacking pallet load units is similar for all analyzed instances. It is 0.7–2.6 s. for 8–20 PLUs per instance, except 51.2 s. for 20 PLUs, see instance 5.

## 4. Conclusions

### 4.1. Summary of the Results Obtained

The research carried out on the construction of stacked palletized load units SPLU resulted in the development of a mathematical model for this decision problem. While the phenomenon of the construction of palletized load units itself is a problem that has been studied for over 50 years, the authors of the publication have recently extended the existing state of knowledge in the direction of off-line type algorithms and the consideration of constraints strongly related to the specifics of the logistics industry, where these problems occur in daily operations. This article deals with the latter trend. Indeed, the authors focused on the phenomenon of PLU stacking in a situation where the process of picking pallet load units for the customer, and more broadly for the ordering party, produces multi-product units that significantly deviate from the permissible height and weight parameters

of pallet units. In addition, there is a high probability that some of the PLUs built in this way will be characterized by a lack of susceptibility to stacking in the process of SPLU construction. This results directly from the fact that there is not enough *top*-type PLU layer support area for the construction of subsequent layers, i.e., subsequent PLUs.

As indicated in the section on the state-of-the-art review, the phenomenon of PLU stacking itself is not a new issue. However, it concerns the situation of parallel PLUs construction planning and vehicle loading planning. The authors of this article addressed the issue of PLU stacking planning in a broader context and as a stand-alone decision problem. Indeed, based on practical experience in the logistics industry, it is noticeable that the need to build SPLUs is not only related to the stacking of load carriers in the vehicle cargo space but also applies to the construction of SPLUs that are created well in advance of the loading process, and therefore, must be subject to periodic storage in racking slots or other spaces with more stringent height parameters compared to the vehicle cargo space. For this reason, the authors decided to develop a mathematical model to meet the requirements of the wider logistics process.

The summary of the research presented in this article, i.e., a comparison of methodological similarities and differences between the classical bin packing problem (BPP) and the proposed methodology of stacking pallet load units problem (SPLUP), is shown in Table 5. The main conclusion is that the SPLUP considered by the authors is different from typical BPP, even combined problem of bin packing and vehicle loading. The mathematical model of SPLUP includes typical BPP constraints and additional restrictions such as filling the top layer, fragility, and height homogeneity. Due to the limited number of similar works, a comparison of different approaches with the perspective of computational experiments, which would indicate the effectiveness of the proposed approach in relation to other studies, is not included. However, in principle, this was not the aim of this work at this stage of research. Finally, the verification of the proposed methodology for the SPLUP solution is limited to the experimental computations of analyzed real-world cases.

The computational experiments illustrated the correctness of the developed model as they show the potential for reducing the space occupied in logistic processes, including the necessary space on the vehicle load bed or in the racking slot of the distribution warehouse. The research focused neither on a large number of instances analyzed nor on orders in which a large number of PLUs created that need to be stacked. This will constitute the subsequent step of the research. The computation time for the analyzed instances and using evolutionary engines was no more than several seconds, except one instance with computation time of around 1 min., which constitutes the entire procedure promising.

**Table 5.** Comparison of methodological similarities and differences between classical bin packing problem and stacking pallet load units problem.

| Comparative Parameter | | Literature [2] | | Presented Research [2] |
|---|---|---|---|---|
| The subject of decision problem | Packaging of products | All [1] exc [3,4,16] | + | - |
| | Packaging of pallet load units | - | − | + |
| | Packaging of products and vehicle loading | [3,4,16] | + | - |
| Type of decision problem | NP-hard type | All [1] | + | + |
| Dimensions of analysis | 1D | [15] | + | - |
| | 2D | [2,7] | + | + |
| | 3D | [1–3,6,12–14] | + | - |
| Item shape regularity | Regular shape | All[1] exc [13] | + | + |
| | Irregular shape | [13] | + | - |

**Table 5.** *Cont.*

| Comparative Parameter | | Literature [2] | | Presented Research [2] |
|---|---|---|---|---|
| Spectrum of applied constraints | Weight of items | All [1] | + | + |
| | Height of items | All [1] | + | + |
| | Vertical support/Filling the layer | [11] | + | + |
| | Load bearing/Fragility | [11,12] | + | + |
| | Stability | [12] | + | - |
| | Height homogeneity | [12] | + | + |
| | Compatibility of items | [17] | + | + |
| | Sequencing | [11] | + | + |
| Dependence on time | On-line procedure | [6,10] | + | - |
| | Off-line procedure | All [1] exc [6,10] | + | + |
| Comparison of different approaches/methods | Methodological comparison | [2,6,9] | + | - |
| | Comparison of CPU time | [4,11,15,16] | + | - |
| Number of items per instance | Hard case (200−1000) | [11,15,17] | + | - |
| | Soft case (up to 200) | All [1] exc [11,15,16] | + | + |

[1] all the reference articles, see references; [2] (+) means the comparative parameter is present, (−) otherwise.

*4.2. Directions for Further Research*

Based on the results obtained and the research assumptions made, the authors of this article plan to develop the research thread undertaken. The following studies are proposed:

- Conducting analytical tests for a wider range of test sets, both regarding the number of PLUs to be stacked within a single order, as well as the number of instances included in the experiments. The orders to be analyzed will also be selected so as to examine a bigger number of *top* and *base* PLUs as well as a different weight and height structure of PLUs, i.e., relatively high and light PLUs vs. low and heavy PLUs.
- Conducting a series of experiments assuming different ranges of parameters of acceptable weight and height of SPLUs built.
- Developing a procedure to support purchasing decisi ons, in which, based on the analysis of the result of the PLU stacking planning process, a possible slight modification of the order size will be indicated. The goal is to determine the benefits of eliminating or significantly reducing the number of *top* type PLUs in the planning process and thus assess the possibility of further reducing the number of SPLUs

**Author Contributions:** Conceptualization, P.S. and H.S.; methodology, P.S.; software, P.S.; validation, H.S.; formal analysis, H.S.; investigation, P.S.; resources, P.S.; data curation, P.S.; writing—original draft preparation, P.S. and H.S.; writing—review and editing, P.S. and H.S.; visualization, P.S.; supervision, H.S.; project administration, P.S.; funding acquisition, P.S. and H.S. All authors have read and agreed to the published version of the manuscript.

**Funding:** This research was funded by the Ministry of Science and Higher Education, Republic of Poland, and was performed at Poznan University of Technology, Faculty of Civil and Transport Engineering, grant number 0416/SBAD/0004.

**Institutional Review Board Statement:** Not applicable.

**Informed Consent Statement:** Not applicable.

**Data Availability Statement:** The data suplemental to this research is attached to the article as Appendix A, see Tables A1–A3.

**Acknowledgments:** The authors of the article are grateful to Frontline Systems Inc. for providing the Analytic Solver software and Solver Engines for research purposes.

**Conflicts of Interest:** The authors declare no conflict of interest.

## Appendix A

This section contain data supplemental to the research, see Tables A1–A3, necessary to reproduce all the experiments that results are presented in Section 3 of this article.

**Table A1.** The data applied to the experimental part of the research on SPLUP, instances 1–5.

| Instance | Parameters [1] | | | | | Instance | Parameters [1] | | | | |
|---|---|---|---|---|---|---|---|---|---|---|---|
| | $i$ | $w_i$ | $h_i$ | $f_i$ | $t_i$ | | $i$ | $w_i$ | $h_i$ | $f_i$ | $t_i$ |
| 1 | 1 | 227 | 372 | 2 | 0 | 4 | 1 | 245 | 414 | 1 | 0 |
| | 2 | 240 | 392 | 2 | 0 | | 2 | 347 | 516 | 3 | 1 |
| | 3 | 233 | 381 | 2 | 0 | | 3 | 260 | 429 | 1 | 0 |
| | 4 | 240 | 392 | 2 | 0 | | 4 | 329 | 498 | 3 | 0 |
| | 5 | 256 | 416 | 3 | 1 | | 5 | 276 | 445 | 2 | 0 |
| | 6 | 186 | 311 | 1 | 0 | | 6 | 291 | 460 | 2 | 0 |
| | 7 | 214 | 353 | 2 | 0 | | 7 | 349 | 518 | 3 | 1 |
| | 8 | 219 | 360 | 2 | 0 | | 8 | 297 | 466 | 2 | 0 |
| | 9 | 222 | 365 | 2 | 0 | 5 | 1 | 390 | 520 | 5 | 1 |
| | - | - | - | - | - | | 2 | 197 | 357 | 2 | 0 |
| 2 | 1 | 199 | 331 | 1 | 0 | | 3 | 345 | 481 | 5 | 1 |
| | 2 | 248 | 405 | 3 | 1 | | 4 | 151 | 317 | 1 | 0 |
| | 3 | 197 | 329 | 1 | 0 | | 5 | 224 | 379 | 2 | 0 |
| | 4 | 202 | 336 | 1 | 0 | | 6 | 356 | 491 | 5 | 1 |
| | 5 | 259 | 421 | 3 | 1 | | 7 | 297 | 441 | 4 | 0 |
| | 6 | 233 | 383 | 3 | 0 | | 8 | 130 | 300 | 1 | 0 |
| | 7 | 196 | 327 | 1 | 0 | | 9 | 221 | 377 | 2 | 0 |
| | 8 | 175 | 295 | 1 | 0 | | 10 | 240 | 393 | 3 | 0 |
| | 9 | 221 | 364 | 2 | 0 | | 11 | 244 | 396 | 3 | 0 |
| | - | - | - | - | - | | 12 | 207 | 365 | 2 | 0 |
| 3 | 1 | 360 | 529 | 3 | 1 | | 13 | 234 | 388 | 3 | 0 |
| | 2 | 348 | 517 | 3 | 1 | | 14 | 355 | 490 | 5 | 1 |
| | 3 | 361 | 530 | 3 | 1 | | 15 | 359 | 494 | 5 | 1 |
| | 4 | 282 | 451 | 2 | 0 | | 16 | 271 | 420 | 3 | 0 |
| | 5 | 303 | 472 | 2 | 0 | | 17 | 303 | 446 | 4 | 0 |
| | 6 | 236 | 405 | 1 | 0 | | 18 | 363 | 497 | 5 | 1 |
| | 7 | 290 | 459 | 2 | 0 | | 19 | 227 | 382 | 2 | 0 |
| | 8 | 297 | 466 | 2 | 0 | | 20 | 364 | 498 | 5 | 1 |

[1] Parameters are expressed in: $i$ (-), $w_i$ (kg), $h_i$ (mm), $f_i$ (-), $t_i$ (-).

**Table A2.** The data applied to the experimental part of the research on SPLUP, instance 6.

| Instance | Parameters [1] | | | | | Instance | Parameters [1] | | | | |
|---|---|---|---|---|---|---|---|---|---|---|---|
| | $i$ | $w_i$ | $h_i$ | $f_i$ | $t_i$ | | $i$ | $w_i$ | $h_i$ | $f_i$ | $t_i$ |
| 6 | 1 | 238 | 422 | 4 | 1 | 6 | 11 | 204 | 380 | 2 | 0 |
| | 2 | 244 | 430 | 4 | 1 | | 12 | 201 | 377 | 2 | 0 |
| | 3 | 219 | 399 | 3 | 0 | | 13 | 201 | 377 | 2 | 0 |
| | 4 | 182 | 353 | 1 | 0 | | 14 | 211 | 389 | 2 | 0 |
| | 5 | 230 | 412 | 3 | 0 | | 15 | 229 | 411 | 3 | 0 |
| | 6 | 208 | 386 | 2 | 0 | | 16 | 229 | 411 | 3 | 0 |
| | 7 | 239 | 424 | 4 | 1 | | 17 | 196 | 370 | 2 | 0 |
| | 8 | 226 | 408 | 3 | 0 | | 18 | 234 | 418 | 3 | 1 |
| | 9 | 224 | 405 | 3 | 0 | | 19 | 246 | 433 | 4 | 1 |
| | 10 | 222 | 402 | 3 | 0 | | 20 | 214 | 393 | 2 | 0 |

**Table A3.** The data applied to the experimental part of the research on SPLUP, instances 7–10.

| Instance | Parameters [1] | | | | | Instance | Parameters [1] | | | | |
|---|---|---|---|---|---|---|---|---|---|---|---|
| | $i$ | $w_i$ | $h_i$ | $f_i$ | $t_i$ | | $i$ | $w_i$ | $h_i$ | $f_i$ | $t_i$ |
| 7 | 1 | 331 | 523 | 5 | 1 | 9 | 1 | 266 | 438 | 3 | 1 |
| | 2 | 197 | 369 | 1 | 0 | | 2 | 146 | 358 | 1 | 0 |
| | 3 | 325 | 516 | 5 | 1 | | 3 | 228 | 413 | 3 | 0 |
| | 4 | 341 | 533 | 5 | 1 | | 4 | 129 | 347 | 1 | 0 |
| | 5 | 308 | 496 | 4 | 1 | | 5 | 219 | 407 | 2 | 0 |
| | 6 | 178 | 347 | 1 | 0 | | 6 | 257 | 432 | 3 | 1 |
| | 7 | 283 | 468 | 4 | 0 | | 7 | 175 | 377 | 2 | 0 |
| | 8 | 232 | 410 | 2 | 0 | | 8 | 211 | 401 | 2 | 0 |
| | 9 | 292 | 477 | 4 | 0 | | 9 | 159 | 367 | 1 | 0 |
| | 10 | 190 | 361 | 1 | 0 | | 10 | 169 | 373 | 1 | 0 |
| | 11 | 175 | 344 | 1 | 0 | | 11 | 179 | 380 | 2 | 0 |
| | 12 | 297 | 483 | 4 | 0 | | 12 | 138 | 352 | 1 | 0 |
| | 13 | 267 | 449 | 3 | 0 | | 13 | 195 | 391 | 2 | 0 |
| | 14 | 237 | 415 | 2 | 0 | | 14 | 181 | 381 | 2 | 0 |
| | 15 | 194 | 365 | 1 | 0 | | 15 | 130 | 347 | 1 | 0 |
| | 16 | 228 | 405 | 2 | 0 | | - | - | - | - | - |
| | 17 | 247 | 427 | 3 | 0 | | - | - | - | - | - |
| | 18 | 300 | 486 | 4 | 1 | | - | - | - | - | - |
| | 19 | 209 | 383 | 1 | 0 | 10 | 1 | 261 | 552 | 4 | 1 |
| | 20 | 288 | 473 | 4 | 0 | | 2 | 231 | 463 | 1 | 0 |
| 8 | 1 | 273 | 442 | 2 | 1 | | 3 | 225 | 444 | 1 | 0 |
| | 2 | 222 | 408 | 2 | 0 | | 4 | 238 | 483 | 2 | 0 |
| | 3 | 152 | 359 | 1 | 0 | | 5 | 232 | 467 | 1 | 0 |
| | 4 | 154 | 363 | 1 | 0 | | 6 | 228 | 455 | 1 | 0 |
| | 5 | 189 | 386 | 1 | 0 | | 7 | 235 | 474 | 2 | 0 |
| | 6 | 148 | 361 | 1 | 0 | | 8 | 227 | 450 | 1 | 0 |
| | 7 | 216 | 404 | 2 | 0 | | 9 | 248 | 513 | 2 | 0 |
| | 8 | 234 | 416 | 2 | 0 | | 10 | 267 | 571 | 5 | 1 |
| | 9 | 214 | 403 | 2 | 0 | | 11 | 226 | 447 | 1 | 0 |
| | 10 | 166 | 371 | 1 | 0 | | 12 | 255 | 534 | 4 | 0 |
| | 11 | 166 | 371 | 1 | 0 | | 13 | 228 | 454 | 1 | 0 |
| | 12 | 155 | 364 | 1 | 0 | | 14 | 226 | 447 | 1 | 0 |
| | 13 | 155 | 364 | 2 | 0 | | 15 | 267 | 570 | 5 | 1 |
| | 14 | 252 | 428 | 2 | 1 | | 16 | 260 | 550 | 4 | 1 |
| | 15 | 212 | 402 | 2 | 1 | | 17 | 265 | 565 | 5 | 1 |
| | 16 | 237 | 418 | 2 | 1 | | 18 | 266 | 568 | 5 | 1 |
| | 17 | 256 | 431 | 2 | 1 | | 19 | 265 | 564 | 5 | 1 |

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
