# Peer review of "Design Optimization of Stacked Pallet Load Units"

_applsci, doi:10.3390/app13042153_

Round 1

Reviewer 1 Report

The submitted paper is devoted to an important problem for rationalization of using space in warehouses by optimization of designing stacked pallet load units. The abstract fully highlights a main aspect considered in the presented research.

After manuscript reading, the next suggestions can be made:

1) The literature review can be improved, because after presented analysing the current state of establishing tasks, I didn't find any thoughts in paper about how previously resolved the problem of designing stacked pallet load units to consider next: cargo quality; physic and chemical characteristics of the product; type of unification packing of each unit located into pallet; and other aspects;

2) Line 247, fragility class have code “l”, but the same parameter denotes as “fi” in Table 3 and equation 7. Please, use only one label for the fragility class, or explain these differences;

3) In my opinion, the main issue of the presented research is connected with the proposed model. It is not entirely clear, how does fragility class influence on final stacking height? Why does the model not consider maximum load on lower layers with boxes on initial (first) pallets in stack, and how does this, again, impact on height?;

4) It will be good to highlight a separate section from the results for section “Discussions”;

5) Please see, what type of template was used (Journal Energies).

Finally, the research results have a sufficient practice aspect for rationalized warehousing logistics in pallet packing aspects and designing stacked pallet load units. Results of research can be accepted to publication after revisions.

Reviewer 2 Report

The manuscript proposes a single-criteria binary programming model to minimize the pallet spaces required to accommodate the constructed units by considering their geometry (dimensions) and weight. The model was tested with an experimental study. The Bin packing problem (BPP) is an old problem as discussed in the introduction. Hence it is required to do a comparative assessment with existing solutions. Some references are here on recent algorithms for bin packing and collison free placement of bins in their positions.

El-Ashmawi, Walaa H., and Diaa Salama Abd Elminaam. "A modified squirrel search algorithm based on improved best fit heuristic and operator strategy for bin packing problem." Applied Soft Computing 82 (2019): 105565.

Kumar, Gulivindala Anil, et al. "A novel Geometric feasibility method to perform assembly sequence planning through oblique orientations." Engineering Science and Technology, an International Journal 26 (2022): 100994.

Santos, Luiz FO Moura, et al. "A variable neighborhood search algorithm for the bin packing problem with compatible categories." Expert Systems with Applications 124 (2019): 209-225.

Section 2 is well organised with the symbols, constraints and mathematical objective functions, however it lacks with implementation schema. Please expand/elaborate how optimization takes place?

As stated it is recommended to compare the results presented in the section 3 with recent literature to draw the merits of the current paper.
